# Characteristics and incidence trends of adults hospitalized with community-acquired pneumonia in Portugal, pre-pandemic

**Joana Carneiro**[1], **Rita Teixeira**[2], **Andreia Leite**[3,4], **Maria Lahuerta**[5], **Julie Catusse**[5], **Mohammad Ali**[5], **Sílvia Lopes**[3]*

1 NOVA University Lisbon, NOVA National School of Public Health, Lisbon, Portugal, 2 Laboratórios Pfizer, Vaccines, Porto Salvo - Lisbon, Portugal, 3 NOVA University Lisbon, NOVA National School of Public Health, CHRC, REAL, CCAL, Lisbon, Portugal, 4 Current affiliation: National Institute of Health Doutor Ricardo Jorge, Department of Epidemiology, Lisbon, Portugal, 5 Global Respiratory Vaccines, Pfizer Inc, Pennsylvania, United States of America

* silvia.lopes@ensp.unl.pt (SL)

## Abstract

Community-acquired pneumonia (CAP) is a major cause of hospitalization that leads to substantial morbidity, mortality, and costs. Evaluating CAP trends over time is important to understand patterns and the impact of public health interventions. This study aims to describe the characteristics and trends in the incidence of adults hospitalized with CAP in Portugal between 2010 and 2018. In this study, we included hospitalization data, prevalence of comorbidities, and population data. CAP hospitalizations of adults (≥18y) living in mainland Portugal discharged from public hospitals were identified using ICD-9-CM or ICD-10-CM codes. Based on previous CAP studies, we selected nine relevant comorbidities. We described the frequency and incidence of CAP hospitalizations per sex, age group, comorbidity, and year of discharge. Trends were explored using Joinpoint regression. We observed 470,545 CAP hospitalizations falling into the 2010–18 period. The majority were males (54.8%) and aged ≥75 years (65.3%). Most often recorded comorbidities were congestive heart failure (26.4%), diabetes (25.5%), and chronic pulmonary disease (19.2%). The Joinpoint regression identified a gradual decline in the incidence rates of CAP hospitalizations for both sexes and all age groups. Of the nine comorbidities selected, seven showed a progressive increase in incidence rates followed by a subsequent decline (all except HIV/AIDS and chronic renal disease). Our findings offer valuable insights for selecting priority groups for public health interventions and design strategies to mitigate the burden of CAP.

**Data availability statement:** As stated in the formal agreement signed with the data provider, Administração Central do Sistema de Saúde (ACSS), the authors are not authorized to share the data publicly. Others may request access

to these data subject to approval by the ACSS (geral@acss.min-saude.pt). The authors confirm that others would be able to access these data in the same manner as the authors and that the authors did not have any special access privileges that others would not have.

**Funding:** This study was conducted as a research collaboration between the NOVA National School of Public Health (sponsor) and Pfizer (funder). The sponsor played a role in the study design, data collection, analysis and interpretation, decision to publish, and preparation of the manuscript. The funder played a role in the study design, data interpretation, decision to publish, and preparation of the manuscript. RT, ML, JuC, and MA are employees of Pfizer and may hold stocks or stock options. Grant number: not applicable. Pfizer URL: https://www.pfizer.com/.

**Competing interests:** I have read the journal's policy and the authors of this manuscript have the following competing interests: JoC is a researcher at NOVA National School of Public Health working on a project funded by Pfizer, Inc. RT, ML, JuC, and MA are employees of Pfizer, Inc. and may hold stocks or stock options. AL coordinates a project funded by Pfizer, Inc. and is employee of NOVA National School of Public Health. SL coordinates a project funded by Pfizer, Inc. and is employee of NOVA National School of Public Health.

**Abbreviations list: APC** – annual percent change; **CAP** – Community-acquired pneumonia; **ECI** – Elixhauser Comorbidity Index; **HIV/AIDS** – Human immunodeficiency virus/ Acquired immunodeficiency syndrome; **ICD**-9-CM – International Classification of Diseases – 9th version – Clinical Modification; **ICD**-10-CM – International Classification of Diseases – 10th version – Clinical Modification; **POA** – Present on admission

## Introduction

Globally, community-acquired pneumonia (CAP) is associated with high mortality, morbidity, and costs for the healthcare system.[1] According to the World Health Organization Global Health Estimates, lower respiratory tract infections, which include pneumonia, were the leading cause of mortality within communicable diseases and the fourth-leading cause overall in 2019.[2] In 2017 in the European Union, pneumonia was the second most often reported cause of death amongst infectious respiratory diseases and accounted for 6% of all deaths from treatable causes in that year.[3] In 2015 Portugal had the highest age-standardized rate of pneumonia mortality amongst 28 European countries, with 57.7 deaths per 100,000 population.[4]

Cases of CAP are generally treated as outpatient, but severe cases may require hospitalization.[5,6] Data on the trends of CAP in Europe is limited, despite its substantial impact on public health. Some studies in different countries identified an increase in CAP hospitalizations among adults between the late 1990s and 2005.[7] A 2018 literature review showed that the overall incidence rate for CAP in Europe varied between 68–7000 cases per 100,000 population in each country and the incidence of CAP hospitalizations between 16–3581 per 100,000.[8] In Portugal, the average annual rate of hospitalizations for adults due to CAP was 3.61 per 1000 population in 2000–09.[8] Another more recent study in Portugal found that hospitalization rates fell slightly between 2012 and 2014, but in-hospital mortality increased, mainly related to ageing and socioeconomic deprivation.[9]

In Portugal, pneumonia represents a substantial share of ambulatory care sensitive conditions – hospitalizations that could have been avoided by adequate prevention, management, treatment, and interventions delivered in the outpatient settings.[10] Moreover, factors such as an ageing population, rising antibiotic resistance, and new pathogens suggest that the trends in pneumonia incidence are likely to worsen in the upcoming years.[11] Having information on existing trends will thus support monitoring of CAP burden. This highlights the significance of our study, as there is a gap in more recent information about the incidence of CAP hospitalizations in Portugal.

Having accurate data on CAP before the COVID-19 pandemic is crucial for understanding trends, identifying priority groups for public health interventions, and designing strategies to mitigate the burden of CAP. Our study aims to characterize adults hospitalized with CAP and to describe trends in adult incidence rates of CAP hospitalizations by year, demographic characteristics, and comorbidities between 2010 and 2018. By analysing demographic characteristics, comorbidities, and the trends of CAP hospitalizations, we seek to provide valuable information for public health interventions, contributing to a reduction in the burden of this disease.

## Methods

### Data sources

**CAP hospitalizations.** Data regarding public hospital discharges were provided by the Portuguese Health System Central Administration for the 2010–18 period. This period was selected based on the most updated and complete database available

at the host institution. Data for 2019 was provided but had not yet been finalized, leading to completeness issues and its subsequent exclusion. Data included demographics (sex and age), district of residence, main and additional diagnoses [coded with International Classification of Diseases – 9th version – Clinical Modification (ICD-9-CM) or with International Classification of Diseases – 10th version – Clinical Modification (ICD-10-CM) codes], whether CAP was present on admission (POA (Present on admission); recorded after 2013), and year of discharge. The study received written approval by the National School of Public Health Ethics Commission (ref. 22/2022). Data analysis started on October 23, 2023. All data were fully anonymized before access to ensure confidentiality of participants. Minors were not included in this study.

**CAP hospitalizations incidence rate.** We used the above-mentioned data to estimate CAP hospitalizations incidence rate. For the denominators, we used various sources. To compute global, sex-specific, and age-specific rates we used the annual number of inhabitants in mainland Portugal, per sex and age group, retrieved from Statistics Portugal[12]; for rates by comorbidity we sought to identify the number of people living with each comorbidity in Portugal from a period overlapping (as much as possible) the period with hospitalizations data (2010–18). We identified these data from the National Health Survey[13] (chronic pulmonary disease, complicated and uncomplicated diabetes, and chronic renal disease; periods: 2005/2006, 2014, and 2019), Global Burden of Disease study[14] (peripheral vascular disease, rheumatoid arthritis/collagen vascular diseases, HIV/AIDS (Human immunodeficiency virus/ Acquired immunodeficiency syndrome), liver disease, and solid tumor without metastases and metastatic tumor; periods: years from 2010 until 2018), and EPICA study[15,16] (congestive heart failure; periods: years 2011 and 2018). Since annual data on comorbidities were not available from the National Health Survey or EPICA studies for the whole study period, we estimated the values for the missing years assuming a linear evolution of the number of people living with the comorbidity between the years with published data.

## Population

Adults (≥18 years old) discharged with a CAP diagnosis from public hospitals in mainland Portugal between 2010 and 2018 were included in the analysis. In Portugal, public hospitals account for the majority of hospitalizations. In 2018, public and public-private partnership hospitals accounted for 83% of the hospitalizations of people aged 65 or older in mainland[17]. Based on previous studies[18,19], we used ICD-9-CM and ICD-10-CM codes to identify CAP hospitalizations (see S1 Table in Supplemental Material). We identified 505,403 hospitalizations that had a CAP code recorded in main or additional diagnoses. Of these, we excluded 34,858 hospitalizations for which the CAP code was not recorded as present on admission (for years 2013 on, POA not available in 2010–12).

## Variables

Demographic variables included sex and age group (18–29, 30–49, 50–64, 65–74, 75–84, and ≥85 years of age). District of residence was categorized according to the 18 districts of mainland Portugal. Comorbidities were studied using the Elixhauser Comorbidity Index (ECI). We selected those that were more important based on previous studies: chronic pulmonary disease, peripheral vascular disease, congestive heart failure, complicated and uncomplicated diabetes, rheumatoid arthritis/collagen vascular diseases, HIV/AIDS, liver disease, chronic renal disease, solid tumor without metastases, and metastatic tumor[19–23]. In order to increase consistency between 9th and 10th versions of ICD codes, we followed the Enhanced ICD-9-CM coding algorithms from Quan et al[24] (see S2 Table in Supplemental Material). The number of selected comorbidities recorded was also studied (none, 1, 2, 3, 4, ≥5). All reported data were included in the analysis.

## Statistical analysis

We conducted a descriptive analysis and utilized joinpoint analysis[25] to study trends of CAP hospitalizations.

We described the frequency of hospitalizations by sex, age group, year, and district of residence. In addition, hospitalizations were also described according to the number and type of comorbidity.

The incidence of CAP hospitalizations amongst adults was calculated per 1,000 persons using the number of CAP hospitalizations (numerator) divided by the population (denominator), for the overall population, by sex, age group, year and comorbidity. Additionally, 95% confidence intervals were also calculated (epi.conf function in R software). We sought to match the definitions of comorbidity, age group, and residence as much as possible between hospitalizations and prevalence data, but some differences remained, as described in S3 Table in Supplemental Material. The comorbidities "solid tumor without metastasis" and "metastatic cancer" were analysed together due to the availability of prevalence data that registered only "cancer".

A sensitivity analysis on the pneumonia identification was performed to assess possible bias in coding. We added the hospitalizations for which pneumonia was recorded as additional diagnosis and POA was coded as "unknown" or "undetermined" (2013 and thereafter, since POA was not available before).

To test for trends and possible changes across the study period we conducted a Joinpoint analysis using the Joinpoint Regression Software from the United States National Institute of Health[25]. This analysis was stratified by sex, age group, and comorbidities.

Other than the Joinpoint regression, the statistical analyses were performed using R 4.1.1.[26].

## Results

In the 2010–18 period, 470,545 hospitalizations with a diagnosis of CAP were identified (Table 1).

Most patients were male (54.8%). Those aged ≥75 years accounted for 65.3% of the hospitalizations (75-84y: 33.4% and ≥ 85y: 31.9%), with the 18-29y being the least-populated group (1.3%). The districts of Lisbon (21.1%) and Porto (12.9%) showed the highest frequency. The most frequently recorded comorbidities were congestive heart failure (26.4%), diabetes (25.5%), and chronic pulmonary disease (19.2%). The majority of individuals hospitalized (65.4%) presented at least one comorbidity, with 8.9% presenting with three or more comorbidities.

The incidence rates and confidence intervals of CAP hospitalizations per 1,000 inhabitants are reported in Table 2, by sex and age group for each year under study. There were 6.6 CAP hospitalizations (95% confidence interval: 6.6-6.7) per 1,000 inhabitants in 2010. This rate peaked at 7.1 (7.1-7.2) in 2012, then presented a lower peak at 6.5 (6.4-6.5) in 2015, before falling to 5.6 (5.5-5.6) in 2018 (Table 2).
The joinpoint analysis showed a gradual decline of CAP hospitalizations incidence rate in the 2010–18 period [annual percent change (APC): -2.04; Fig 1].

In 2018 incidence rates were higher for male sex (6.4, 95%CI: 6.3-6.5), compared to female (4.9, 4.8-4.9; Table 2). A gradual decline was also observed for both, steeper for male sex (APC: -2.5 vs. female: -1.5). Per age group, incidence rate was higher in ≥ 85y (51.9 in 2018, 95%CI: 51.1-52.7) and 75-84y (18.9, 18.6-19.2). There was a gradual fall of CAP hospitalization incidence rate across all age groups, with no changes in trend (Fig 1). This decrease was lowest in the age group with the highest incidence (APC: -2.72 in aged ≥ 85y; -8.82 to -3.97 in remaining age groups).

The comorbidities that presented higher CAP hospitalization incidence rates in 2018 were congestive heart failure (35.8, 95%CI: 35.2-36.4), chronic renal disease (22.0, 21.6-22.5), rheumatoid arthritis/collagen vascular diseases (21.4, 19.9-22.9), and chronic pulmonary disease (18.4, 18.0-18.7; Table 2). Liver disease had the lowest incidence rate (1.0, 0.99-1.08). Of the nine selected comorbidities, seven showed a progressive increase in incidence rates followed by a subsequent decline (all except HIV/AIDS and chronic renal disease; Fig 2). That decline was more accentuated for liver disease (APC: -13.96 in 2016–18) and less for congestive heart failure (-0.64 in 2012–18). Incidence rate of CAP hospitalizations decreased in HIV/AIDS over the study period (APC: -3.64). During the 2012–18 period the incidence rates increased for people living with chronic renal disease (APC: 1.65).

Incidence and admission rates obtained in the sensitivity analysis (see S6 Table and S7 Table in Supplemental Material) were consistent with the main analysis.

**Table 1. Frequency of CAP hospitalizations by patients' characteristics and year.**

|  | Number of hospitalizations (%) |
|---|---|
| **Total** | 470 545 (100.0%) |
| **Sex** |  |
| Male | 257 834 (54.8%) |
| Female | 212 711 (45.2%) |
| **Age** |  |
| 18-29 | 6 257 (1.3%) |
| 30-49 | 28 224 (6.0%) |
| 50-64 | 54 351 (11.6%) |
| 65-74 | 74 440 (15.8%) |
| 75-84 | 157 345 (33.4%) |
| ≥85 | 149 928 (31.9%) |
| Mean (Standard Deviation) | 75.9 (14.8) |
| Median (IQR) | 79.7 (16.9) |
| **Year** |  |
| 2010 | 54 557 (11.6%) |
| 2011 | 56 054 (11.9%) |
| 2012 | 58 424 (12.4%) |
| 2013 | 52 096 (11.1%) |
| 2014 | 50 228 (10.7%) |
| 2015 | 52 895 (11.2%) |
| 2016 | 50 556 (10.7%) |
| 2017 | 49 825 (10.6%) |
| 2018 | 45 910 (9.8%) |
| **District of residence** |  |
| Aveiro | 32 851 (7.0%) |
| Beja | 5 878 (1.2%) |
| Braga | 34 744 (7.4%) |
| Bragança | 9 058 (1.9%) |
| Castelo Branco | 13 460 (2.9%) |
| Coimbra | 32 059 (6.8%) |
| Évora | 5 580 (1.2%) |
| Faro | 19 996 (4.2%) |
| Guarda | 9 216 (2.0%) |
| Leiria | 31 811 (6.8%) |
| Lisboa | 99 203 (21.1%) |
| Portalegre | 7 823 (1.7%) |
| Porto | 60 892 (12.9%) |
| Santarém | 31 458 (6.7%) |
| Setúbal | 32 928 (7.0%) |
| Viana do Castelo | 11 060 (2.4%) |
| Vila Real | 14 136 (3.0%) |
| Viseu | 18 391 (3.9%) |
| **Number of comorbidities** |  |
| None | 162 2935 (34.6%) |
| 1 | 172 392 (36.6%) |
| 2 | 92 978 (19.8%) |

*(Continued)*

**Table 1.** (Continued)

|  | Number of hospitalizations (%) |
| --- | --- |
| 3 | 33 534 (7.1%) |
| 4 | 7 568 (1.6%) |
| ≥5 | 1 140 (0.2%) |
| **Comorbidities** |  |
| Congestive heart failure | 124 339 (26.4%) |
| Diabetes | 119 911 (25.5%) |
| Chronic pulmonary disease | 90 577 (19.2%) |
| Chronic renal disease | 74 386 (15.8%) |
| Solid tumor without metastasis and Metastatic cancer | 37 672 (8.0%) |
| Liver disease | 22 689 (4.8%) |
| Peripheral vascular disorders | 15 375 (3.3%) |
| Rheumatoid arthritis/collagen vascular diseases | 7 971 (1.7%) |
| HIV/AIDS | 2 098 (0.4%) |

AIDS: acquired immunodeficiency syndrome, CAP: community-acquired pneumonia, HIV: human immunodeficiency virus.

## Discussion

Our study presents important data on the characteristics and incidence trends in CAP hospitalizations in Portugal over a nine-year period. With 470,545 CAP hospitalizations included in this study, males and those aged ≥75 years accounted for the largest share of CAP hospitalizations. Many patients suffered from congestive heart failure, diabetes, and chronic pulmonary disease. Incidence rate of CAP hospitalizations per 1,000 adults varied between 7.1 (7.1-7.2) in 2012 and 5.6 (5.5-5.6) in 2018 and fell during the 2010–18 period [annual percent change (APC): -2.04]. A drop was observed for all sex and age subgroups, but was steeper for male patients (APC: -2.5) and those with <85 years (APC: -2–72; other age groups: between -8.82 and -3.97). For seven of the nine comorbidities studied, CAP hospitalization incidence rates rose and then fell (exceptions were HIV/AIDS and chronic renal disease).

The higher frequency of male sex in CAP hospitalizations aligns with findings from other studies that suggest an association with smoking or alcoholism.[7,9] The observed age distribution is consistent with the epidemiological pattern of CAP, whereby older age groups (≥65 years) usually present a higher incidence of CAP compared to younger groups. [9,27] Regarding the higher frequency of congestive heart failure, diabetes, and chronic pulmonary disease, Nguyen et al.[22] also observed that a high proportion of the CAP population lived with congestive heart failure and chronic pulmonary disease. The CAP hospitalization incidence rates in this study were higher than those described in a study in Germany for 2015 (423 per 100,000 person-years of observation; 95%CI: 416–430).[28] However, results for the two sensitivity analyses in which the list of ICD-10 codes to define CAP was expanded (611, 603–620 and 570, 562–579) were more similar to those found in Portugal, even if still lower (year 2015: 6.5 per 1,000 inhabitants, 6.4-6.5). The decreasing trend we observed in 2010–18 contrasts with the growth described for the previous years. A study from Froes et al. showed that the average annual rate of CAP hospitalizations per 1,000 population increased consistently from 3.02 in 2000 to 4.70 in 2009.[18] For the period 2001–11, Pessoa et al. found a rise in the CAP hospitalization rate from 2.8 to 4.3 per 1,000 population in Portugal, but the number of hospitalizations fell slightly after 2012 in people ≥65 years[9]. In our study, for both sexes and some age groups, the overall incidence rate started decreasing from 2012 on, a similar trend to what was observed in the Pessoa et al. study[9]. However, comparisons between studies are limited due to differences in the definition of CAP hospitalization (ICD codes included and their position – main or additional).

**Table 2. Incidence rates of CAP hospitalizations per 1,000 inhabitants, estimated for each year, total and by sex and age group. Incidence rates of CAP hospitalizations per 1,000 adults living with comorbidities, estimated for each year, by comorbidity. Years 2010-18.**

| | Incidence rates (95%CI) per 1,000 inhabitants | | | | | | | | |
|---|---|---|---|---|---|---|---|---|---|
| | 2010 | 2011 | 2012 | 2013 | 2014 | 2015 | 2016 | 2017 | 2018 |
| **Total** | 6.6 (6.6-6.7) | 6.8 (6.7- 6.9) | 7.1 (7.1-7.2) | 6.4 (6.3-6.4) | 6.2 (6.1-6.2) | 6.5 (6.4-6.5) | 6.2 (6.1-6.2) | 6.1 (6.0-6.1) | 5.6 (5.5-5.6) |
| **Sex** | | | | | | | | | |
| Male | 7.9 (7.8- 8.0) | 8.1 (8.0 - 8.2) | 8.3 (8.2 - 8.4) | 7.6 (7.5 - 7.7) | 7.2 (7.1 - 7.3) | 7.5 (7.4 - 7.6) | 7.2 (7.1- 7.3) | 7.1 (7.0-7.1) | 6.4 (6.3 - 6.5) |
| Female | 5.5 (5.4- 5.5) | 5.7 (5.6 - 5.7) | 6.1 (6.0 - 6.1) | 5.3 (5.2 - 5.4) | 5.2 (5.1 - 5.3) | 5.6 (5.5 - 5.7) | 5.3 (5.2- 5.4) | 5.3 (5.2-5.3) | 4.9 (4.8 - 4.9) |
| **Age group (years)** | | | | | | | | | |
| 18-29 | 0.7 (0.6- 0.7) | 0.7 (0.7 - 0.8) | 0.9 (0.9 - 1.0) | 0.4 (0.4 - 0.4) | 0.5 (0.4 - 0.5) | 0.4 (0.4 - 0.5) | 0.4 (0.4- 0.5) | 0.4 (0.3-0.4) | 0.4 (0.3 - 0.4) |
| 30-49 | 1.4 (1.4- 1.4) | 1.4 (1.4 - 1.4) | 1.2 (1.2 - 1.3) | 1.1 (1.1 - 1.1) | 1.1 (1.1 - 1.1) | 1.0 (1.0 - 1.0) | 1.0 (1.0- 1.0) | 0.8 (0.8-0.9) | 0.8 (0.8 - 0.9) |
| 50-64 | 3.4 (3.3- 3.5) | 3.7 (3.6 - 3.7) | 3.3 (3.2 - 3.4) | 3.1 (3.0 - 3.1) | 2.9 (2.8 - 3.0) | 2.9 (2.8 - 2.9) | 3.0 (2.9- 3.0) | 2.7 (2.6-2.7) | 2.5 (2.4 - 2.5) |
| 65-74 | 9.5 (9.3- 9.6) | 9.5 (9.3 - 9.7) | 9.4 (9.2 - 9.5) | 8.0 (7.9 - 8.2) | 7.6 (7.4 - 7.8) | 7.5 (7.4 - 7.7) | 7.1 (7.0- 7.3) | 6.9 (6.7-7.0) | 6.2 (6.1 - 6.3) |
| 75-84 | 27.2 (26.8- 27.6) | 26.6 (26.2 - 27.0) | 27.7 (27.4 - 28.1) | 24.3 (24.0 - 24.7) | 22.6 (22.3 - 22.9) | 24.1 (23.8 - 24.5) | 22.1 (21.8- 22.4) | 21.5 (21.2-21.8) | 18.9 (18.6 - 19.2) |
| ≥85 | 66.6 (65.5- 67.6) | 62.6 (61.6 - 63.6) | 69.3 (69.3 - 70.4) | 62.3 (61.3 - 63.2) | 57.7 (56.8 - 58.6) | 62.0 (61.1 - 62.9) | 56.9 (56.1- 57.8) | 57.5 (56.7-58.4) | 51.9 (51.1 - 52.7) |
| **Comorbidities** | | | | | | | | | |
| Chronic pulmonary disease | 16.3 (15.9-16.6) | 18.3 (17.9-18.6) | 18.0 (17.6-18.4) | 20.0 (19.6-20.4) | 20.4 (20.0-20.8) | 22.7 (22.2-23.1) | 21.4 (21.2-22.0) | 19.0 (18.6-19.3) | 18.4 (18.0-18.7) |
| Congestive heart failure | 33.0 (32.4-33.6) | 34.9 (34.3-35.5) | 39.9 (39.3-40.6) | 36.8 (35.6-36.8) | 35.1 (34.5-35.7) | 38.0 (37.4-38.6) | 36.7 (35.0-37.0) | 38.4 (37.8-38.6) | 35.8 (35.2-36.4) |
| Peripheral vascular disorders | 3.4 (3.3-3.6) | 3.9 (3.7-4.1) | 3.8 (3.6-4.0) | 4.5 (4.3-4.7) | 4.2 (4.0-4.4) | 4.6 (4.4-4.8) | 4.2 (4.0-4.3) | 3.6 (3.4-3.7) | 3.2 (3.0-3.4) |
| Diabetes | 16.7 (16.4-17.0) | 17.0 (16.7-17.3) | 17.2 (17.0-17.5) | 16.7 (16.4-16.9) | 15.9 (15.6-16.1) | 16.7 (16.4-17.0) | 16.2 (15.8-16.3) | 15.9 (15.5-16.0) | 14.5 (14.3-14.8) |
| Rheumatoid arthritis/ collagen vascular diseases | 22.7 (21.1-24.3) | 23.7 (22.1-25.3) | 22.8 (21.2-24.4) | 23.6 (22.1-25.2) | 24.9 (23.3-26.5) | 26.3 (24.7-27.9) | 27.0 (25.1-28.5) | 22.6 (20.6-23.7) | 21.4 (19.9-22.9) |
| HIV/AIDS | 8.2 (7.2-9.3) | 8.4 (7.3-9.4) | 7.9 (6.9-8.9) | 7.9 (6.9-8.8) | 5.6 (4.8-6.4) | 6.7 (5.8-7.5) | 5.8 (4.9-6.4) | 6.8 (5.9-7.6) | 6.7 (5.9-7.5) |
| Solid tumour without metastasis and Mestastatic cancer | 12.6 (12.2-13.0) | 13.2 (12.8-13.6) | 14.2 (13.8-14.6) | 13.6 (13.0-13.8) | 12.8 (12.2-12.9) | 13.9 (13.5-14.3) | 13.3 (12.7-13.5) | 12.9 (12.5-13.3) | 11.1 (10.7-11.4) |
| Liver disease | 1.3 (1.2-1.3) | 1.3 (1.3-1.4) | 1.1 (1.1-1.2) | 1.3 (1.2-1.3) | 1.3 (1.2-1.3) | 1.3 (1.2-1.3) | 1.4 (1.3-1.4) | 1.1 (1.1-1.1) | 1.0 (0.99-1.08) |
| Chronic renal disease | 14.7 (14.3-15.0) | 17.3 (16.9-17.7) | 22.0 (21.6-22.5) | 19.8 (19.3-20.2) | 20.6 (20.2-21.1) | 23.6 (23.1-24.1) | 22.4 (21.7-22.6) | 23.7 (23.2-24.2) | 22.0 (21.6-22.5) |

Note: To compute incidence rates per sex and age group, we considered as numerator the number of CAP hospitalizations and as denominator the number of respective inhabitants (presented in S4 Table in Supplemental Material). To compute incidence rates for comorbidities, we considered as numerator the number of CAP hospitalizations and as denominator the estimated number of people living with each comorbidity (presented in S5 Table in Supplemental Material).

CI: confidence interval

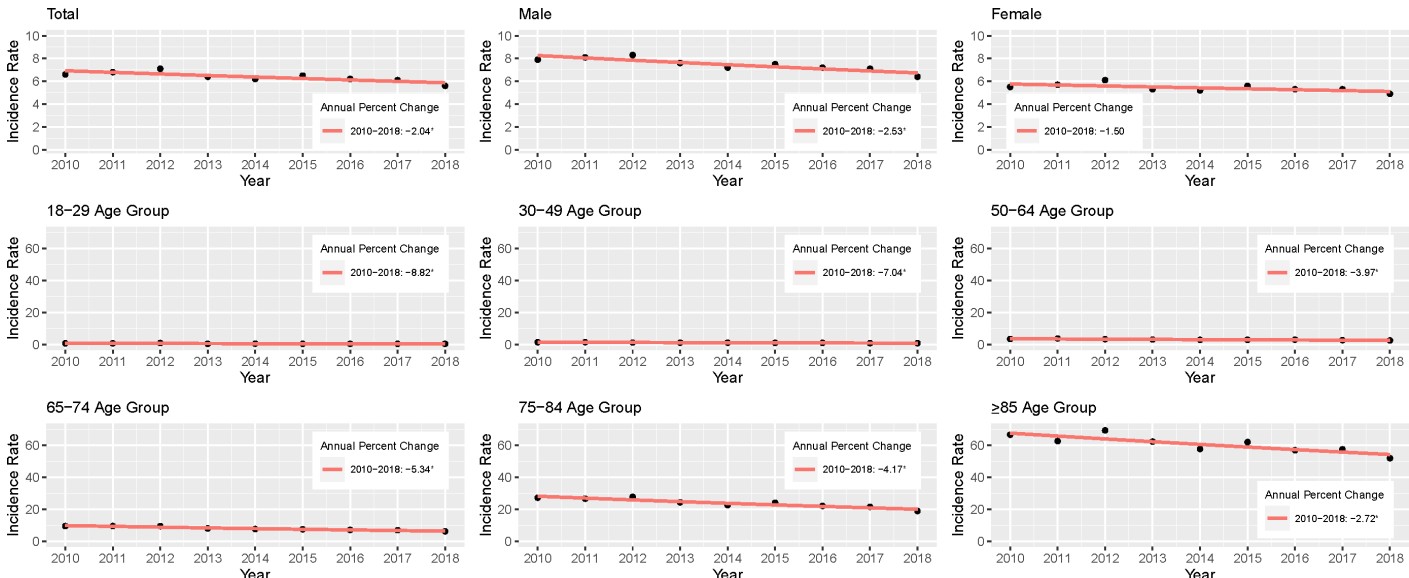

**Fig 1. Joinpoint trend analysis of the incidence rate of CAP hospitalizations, total and by sex and age group, 2010-18.** *The annual percent changes (APC) is significantly different from 0 at alpha = 0.05.

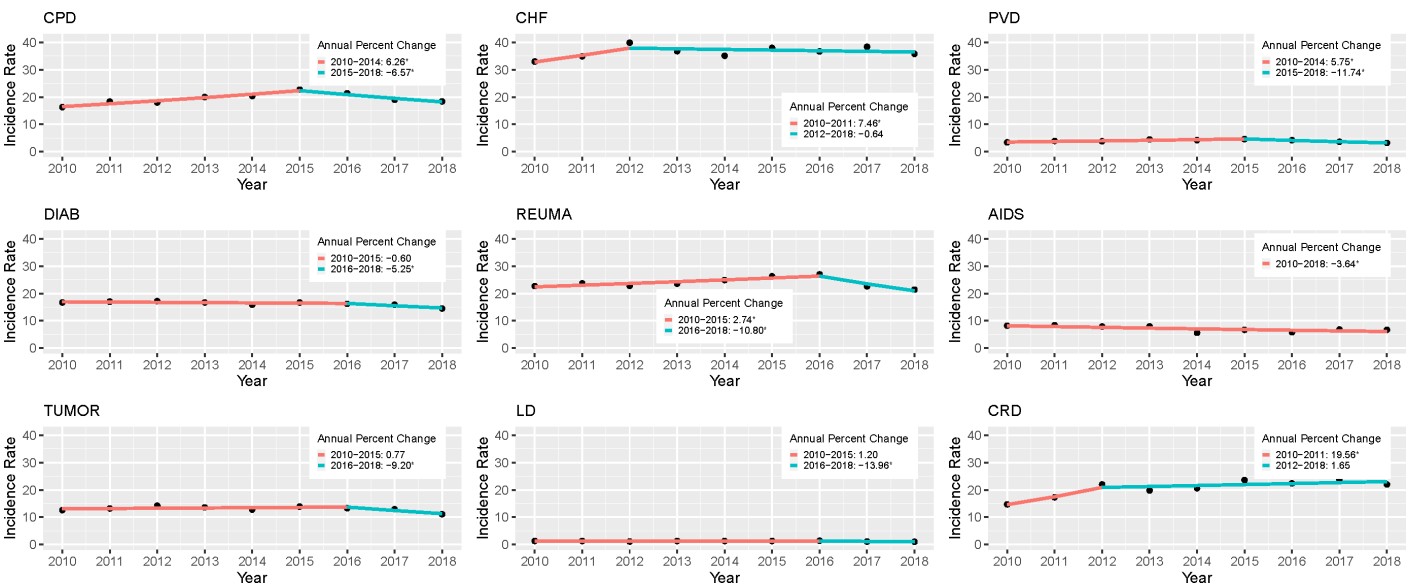

**Fig 2. Joinpoint trend analysis of the incidence rate of CAP hospitalizations, by comorbidity, 2010-18.** CPD: chronic pulmonary disease, CHF: congestive heart failure, PVD: peripheral vascular disease, DIAB: complicated and uncomplicated diabetes, RHEUMA: rheumatoid arthritis/collagen vascular diseases, AIDS: HIV/AIDS, TUMOR: solid tumor without metastases and metastatic tumor, LD: liver disease, CRD: chronic renal disease. *The annual percent changes (APC) is significantly different from 0 at alpha = 0.05.

Despite an overall downward trend in CAP hospitalizations incidence, there were differences in its magnitude and year of start across the population subgroups studied. This may be linked to the fact that hospitalizations are a multifactorial phenomenon, and there are thus several possible explanations for the trends observed. Even though 7-valent pneumococcal

conjugate vaccine was used in the private sector for pediatric immunization from 2001 with an uptake of around 50% each year, in July 2015 13-valent pneumococcal conjugate vaccine was included in the National Immunization Program for all children under 2 years old, in a catch-up program[29] with 98% uptake. Also, during the study period there was a growth in the number of people vaccinated for influenza[30] (58.6% in 2009–10 and 76.0% in 2019–20, for people aged ≥65). Although previous studies have linked vaccines uptake with a decrease of CAP[31,32], this was not specifically addressed in our study. Future studies that link vaccine uptake and hospitalizations data at the individual level, especially for high-risk patients, may shed light on this issue and add to current studies. A decrease in the percentage of CAP cases requiring hospitalization and/or in the total number of CAP cases may reside at the root of reduced hospitalizations. However, we cannot exclude the possibility of a change in hospitalization criteria. In that case, the reduction of hospitalizations could be due to changes in clinical practice that led to a greater share of patients being treated in the outpatient setting.

Our results may be influenced by several limitations. Firstly, when calculating incidence rates per comorbidity, it was not possible to match entirely the definition of the population for which we observed the hospitalizations (numerator) and the number of people living with the condition (denominator), due to differences in age group and/or geographical area in available data. Their effect on incidence rates of each comorbidity is unclear, as these possible biases may have opposite directions and have simultaneously a risk of underestimation and overestimation. Secondly, we also lacked data in specific years for some comorbidities (see S3 Table in Supplemental Material for details) and for the "present on admission" flag (years 2010–12). The latter may have led to an overestimation of incidence rate in those years since it was not possible to distinguish hospital acquired-pneumonia. However, year 2013 was not identified as a turning point, so the effects of not having that flag may be limited. Thirdly, ICD-10-CM was introduced in Portugal in 2017 (see S1 Table in Supplemental Material[18,19]), but the effect of this change also seems to be limited, as the joinpoint analysis did not identify 2017 as a turning point. Fourthly, we were not able to include hospitalization data from private hospitals, which limited our capacity to fully capture the national trends of CAP hospitalizations. So, an increase in its share would lead us to overestimating the decrease of hospitalizations. However, it is unlikely this alone could explain the observed decrease, as the majority of hospitalization in Portugal occur in the public setting[17]. Fifthly, we were not able to include data after 2018, which prevented us from analysing potential differences in CAP hospitalization trends (rise, decrease, or both) during the COVID-19 pandemic and post-pandemic periods. Finally, while the evolution of mortality of hospitalized patients was out of the scope of this study, it is an important aspect from the perspective of patients and healthcare providers that may be considered to describe the burden of CAP in future studies.

To reduce the incidence of CAP hospitalizations, especially in groups for which there was an increasing trend or a slower decrease, it may be necessary to develop outpatient support that avoids hospitalization and/or reduces severe cases of CAP. Our findings underscore the importance of having outpatient diagnosis codes that allow studying both inpatient and outpatient encounters related to CAP. Such coding is now being envisaged in Portugal.[33]

## Conclusions

This study provides valuable insights on the trends of CAP hospitalizations over time using a large dataset of public hospitals and demonstrates the value of real-world data to inform public health interventions. By analyzing the incidence of CAP hospitalizations stratified by factors such as age group, sex or comorbidities our study provides relevant data to identify higher-risk populations. Our study also highlights the importance of having complete, accurate and timely health data from public and private health settings to inform policy making. Further studies are needed to evaluate the subsequent trends of CAP hospitalizations and its causes post COVID-19 to inform the design of strategies to mitigate the burden of community-acquired pneumonia.

## Supporting information

**S1 Table:  List of ICD-9-CM and ICD-10-CM codes.**
(DOCX)

**S2 Table:  ICD-9-CM and ICD-10-CM Coding Algorithms for Elixhauser Comorbidities Index[24].**
(DOCX)

**S3 Table:  Criteria for estimating number of adults living with comorbidities.**
(DOCX)

**S4 Table:  Number of CAP hospitalizations and number of inhabitants, estimated for each year under study, total and by sex and age group, 2010–18.**
(DOCX)

**S5 Table:  Number of CAP hospitalizations and number of adults living with the comorbidity, estimated for each year, by comorbidity, 2010–18.**
(DOCX)

**S6 Table:  Sensitivity analysis: Incidence rates of CAP hospitalizations per 1,000 inhabitants, estimated for each year, by sex and age group, 2013–18.**
(DOCX)

**S7 Table:  Sensitivity analysis: Incidence rates of CAP hospitalizations per 1,000 adults living with the comorbidity, estimated for each year, by comorbidity, 2013–18.**
(DOCX)

## Acknowledgments

We thank the Health System Central Administration for providing the data for this study. We also thank João Victor Rocha for support in the protocol development and Francisco Madeira for helping with 2016 data preparation.

This work is original and has not been published elsewhere, nor is it currently under consideration for publication elsewhere than the European Respiratory Society Congress 2024, where it was presented as a poster.

## Author contributions

**Conceptualization:** Rita Teixeira, Andreia Leite, Sílvia Lopes.

**Data curation:** Joana Carneiro, Sílvia Lopes.

**Formal analysis:** Joana Carneiro.

**Funding acquisition:** Andreia Leite, Sílvia Lopes.

**Methodology:** Joana Carneiro, Rita Teixeira, Andreia Leite, Maria Lahuerta, Mohammad Ali, Sílvia Lopes.

**Project administration:** Sílvia Lopes.

**Resources:** Sílvia Lopes.

**Software:** Joana Carneiro.

**Supervision:** Andreia Leite, Sílvia Lopes.

**Validation:** Rita Teixeira, Andreia Leite, Maria Lahuerta, Julie Catusse, Mohammad Ali, Sílvia Lopes.

**Writing – original draft:** Joana Carneiro, Sílvia Lopes.

**Writing – review & editing:** Rita Teixeira, Andreia Leite, Maria Lahuerta, Julie Catusse, Mohammad Ali, Sílvia Lopes.

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
