## [Decision Letter · Decision Letter 0]

17 Jan 2025

PONE-D-24-56883Characteristics and incidence trends of adults hospitalized with community-acquired pneumonia in Portugal, pre-pandemicPLOS ONE

Dear Dr. Carneiro,

Thank you for submitting your manuscript to PLOS ONE. After careful consideration, we feel that it has merit but does not fully meet PLOS ONE’s publication criteria as it currently stands. Therefore, we invite you to submit a revised version of the manuscript that addresses the points raised during the review process. The reviewers pointed out several issues in their comments. I agree with them. Please rewrite the text and justify any suggestions you don't accept.

We look forward to receiving your revised manuscript.

Kind regards,

Alexandre Morais Nunes, Ph.D.

Academic Editor

PLOS ONE

Journal Requirements:

2. Please amend your list of authors on the manuscript to ensure that each author is linked to an affiliation. Authors’ affiliations should reflect the institution where the work was done (if authors moved subsequently, you can also list the new affiliation stating “current affiliation:….” as necessary).’

Reviewers' comments:

Reviewer's Responses to Questions

**Comments to the Author**

1. Is the manuscript technically sound, and do the data support the conclusions?

Reviewer #1: Yes

Reviewer #2: No

2. Has the statistical analysis been performed appropriately and rigorously? 

Reviewer #1: Yes

Reviewer #2: No

3. Have the authors made all data underlying the findings in their manuscript fully available?

Reviewer #1: Yes

Reviewer #2: No

4. Is the manuscript presented in an intelligible fashion and written in standard English?

Reviewer #1: Yes

Reviewer #2: Yes

5. Review Comments to the Author

Reviewer #1: The paper presents an interesting and relevant topic. It's a very interesting, complete and up-to-date articleHowever, it suffers from several drawbacks that need to be corrected before its possible acceptance.

- Introduction is very very short. I think a little more literature review would be interesting to better understand the problem under study.

- Methods chapter needs more specification of empirical work: Why 2010-18 period? and not other? How did you work with the data? Did you use statistical methods?

- The results are very well organized and are always complemented by tables and explanations of the main highlights.

- The discussion is complet, but the conclusion is short. The conclusion says:"This study provides valuable insights for selecting priority groups in future public health270 interventions". Can you explain a little more?

Congrats!

Reviewer #2: One significant limitation of the study is the exclusion of data from private hospitals, which could potentially affect the generalisability of the findings. If private hospitals experienced an increase in CAP hospitalisations during the study period, this could lead to an overestimation of the decrease in public hospitalisations, thereby distorting the overall trend of CAP incidence. Furthermore, the absence of private hospital data precludes a comprehensive understanding of the national hospitalisation trends. To what extent could the exclusion of private hospitals, where CAP hospitalisation rates may have differed from the public sector, influence the observed downward trend in CAP hospitalizations? How might the inclusion of private hospital data alter the study’s conclusions about the national burden of CAP?

The study encountered difficulties in aligning the hospitalisation data (numerator) with the total population affected by specific comorbidities (denominator). This mismatch could lead to either underestimations or overestimations in calculating the incidence rates of CAP associated with comorbidities. Such discrepancies may limit the precision of the study's estimates and its interpretation of the role of comorbid conditions in CAP hospitalizations. How would a more precise matching of the populations affected by comorbidities with the hospital admission data influence the observed incidence rates? What impact could this mismatch have on the interpretation of trends related to comorbid conditions such as diabetes, congestive heart failure, or chronic pulmonary disease?

The absence of key data for certain years, especially the "present on admission" flag for the 2010-2012 period, introduces a potential bias in the estimation of CAP incidence. Without this flag, distinguishing between community-acquired and hospital-acquired pneumonia becomes impossible, which may lead to an overestimation of CAP-related hospitalizations. This limitation is particularly relevant for early years in the study, where the lack of this data could have inflated the reported incidence rates of CAP. To what extent does the absence of the "present on admission" flag during 2010-2012 affect the accuracy of the reported incidence rates for CAP hospitalizations? Could this limitation significantly impact the overall trend analysis, especially in the earlier years of the study?

The transition to the ICD-10-CM coding system in 2017 is another potential source of bias. While the study posits that the impact of this change was minimal, it is possible that the reclassification of cases under the new system could have affected the trend analysis, particularly if the coding criteria for CAP hospitalisations were altered. The lack of a comprehensive examination of this transition could limit the study's ability to account for variations in coding practices over time. How did the introduction of the ICD-10-CM coding system in 2017 influence the classification of CAP hospitalizations? Could this change have impacted the observed trends in CAP incidence, and to what degree should future studies adjust for potential shifts in coding practices?

Another notable limitation is the lack of data extending beyond 2018, preventing an analysis of CAP hospitalisation trends during and after the COVID-19 pandemic. Given that the pandemic has had a profound impact on healthcare systems worldwide, including shifts in the incidence and management of respiratory diseases such as CAP, the absence of post-2018 data leaves a significant gap in understanding the full impact of the pandemic on CAP hospitalisation trends. How did the COVID-19 pandemic influence CAP hospitalisation trends in Portugal, and what changes can be observed in the post-pandemic period? Would the observed trends in this study continue to hold in the context of the pandemic, or are they likely to be significantly altered?

The study does not address the mortality rates of hospitalised CAP patients, an important aspect of the disease burden. Mortality data would provide a more nuanced understanding of the severity of CAP, especially in relation to comorbidities, and could help determine the effectiveness of healthcare interventions aimed at reducing severe outcomes. The absence of this data limits the study's ability to assess the full clinical impact of CAP hospitalizations. How did mortality rates among hospitalised CAP patients evolve over the study period, and how might these rates relate to the observed reduction in CAP hospitalizations? Would the inclusion of mortality data provide a more comprehensive assessment of the impact of CAP and inform future public health strategies?

The study mentions the increase in pneumococcal and influenza vaccination uptake as a potential factor in the observed decrease in CAP hospitalisations but does not directly link these interventions to the hospitalisation trends. The study acknowledges the influence of vaccination but does not explore this relationship in depth, leaving a gap in understanding the role of vaccines in reducing the incidence of CAP. How did the increased uptake of pneumococcal and influenza vaccinations contribute to the observed decline in CAP hospitalizations? Would a more detailed analysis of vaccination data at the individual level help clarify the role of these public health interventions in reducing the burden of CAP?

6. PLOS authors have the option to publish the peer review history of their article (what does this mean?). If published, this will include your full peer review and any attached files.

Reviewer #1: **Yes: **Andreia Matos

Reviewer #2: **Yes: **Ricardo de Moraes e Soares

---

## [Author Response · Author response to Decision Letter 1]

10 Feb 2025

Response to comments from editor

Thank you for submitting your manuscript to PLOS ONE. After careful consideration, we feel that it has merit but does not fully meet PLOS ONE’s publication criteria as it currently stands. Therefore, we invite you to submit a revised version of the manuscript that addresses the points raised during the review process.

We thank the Editor and the Reviewers for the time dedicated to review our work and the opportunity to revise the manuscript. We acknowledge the caveats of the study and the inherent limitations with the use of real-world data. Following the reviewer´s comments, we have provided additional information and description of the limitations in the discussion. We believe our analysis reports important public health data about the burden of community-acquired pneumonia in Portugal, for a large population (more than 470 thousand hospitalizations) during 9 years, based on a generally used database [1–4] and adds an innovative analysis by including trends by comorbidities.

Please find below the description of actions taken in reply to each of the reviewers’ comments. We have adressed the comments into the new version of the manuscript that has been re-submitted. We are confident that the revised version of our manuscript is of interest to readers and meets PLOS ONE’s publication criteria.

Response to journal requirements

We reviewed and updated the manuscript to ensure it meets PLOS ONE's style requirements, these changes are not tracked for clarity purposes.

Please amend your list of authors on the manuscript to ensure that each author is linked to an affiliation. Authors’ affiliations should reflect the institution where the work was done (if authors moved subsequently, you can also list the new affiliation stating “current affiliation:….” as necessary).’

The authors’ affiliations have been amended.

Please include captions for your Supporting Information files at the end of your manuscript, and update any in-text citations to match accordingly.

We added captions for supporting information at the end of the manuscript and revised the text accordingly.

We reviewed the reference list as requested. We also added five references, due to changes following suggestions from Reviewers 1 and 2.

Response to comments from Reviewer 1

The paper presents an interesting and relevant topic. It's a very interesting, complete and up-to-date article. However, it suffers from several drawbacks that need to be corrected before its possible acceptance.

We thank the Reviewer for the careful review of our manuscript and for the constructive feedback.

Introduction is very very short. I think a little more literature review would be interesting to better understand the problem under study.

This is a fair point. As suggested, we added the sentences below to the introduction, aiming to provide additional relevant information to readers and strengthen the importance of our study. These changes are tracked in the revised manuscript submitted.

“In Portugal, pneumonia represents a substantial share of ambulatory care sensitive conditions – hospitalizations that could have been avoided by adequate prevention, management, treatment, and interventions delivered in the outpatient settings.[10] Moreover, factors such as an ageing population, rising antibiotic resistance, and new pathogens suggest that the trends in pneumonia incidence are likely to worsen in the upcoming years.[11] Having information on existing trends will thus support monitoring of CAP burden.”

“Having accurate data on community-acquired pneumonia (CAP) before the COVID-19 pandemic is crucial for understanding trends, identifying priority groups for public health interventions, and designing strategies to mitigate the burden of CAP.”

Methods chapter needs more specification of empirical work: Why 2010-18 period? and not other? How did you work with the data? Did you use statistical methods?

We followed the Reviewer’s suggestion and added the justification of the period in the methods section (in “CAP hospitalizations”). We also added the sentence below to the beginning of the description of “Statistical analysis”.

“This period was selected based on the most updated and complete database available at the host institution. Data for 2019 was provided but had not yet been finalized, leading to completeness issues and its subsequent exclusion.”

“We conducted a descriptive analysis and utilized joinpoint analysis [25] to study trends of CAP hospitalizations.”

The results are very well organized and are always complemented by tables and explanations of the main highlights.

Thank you for your positive comment on our work.

The discussion is complet, but the conclusion is short. The conclusion says:"This study provides valuable insights for selecting priority groups in future public health270 interventions". Can you explain a little more?

Following the Reviewer’s suggestion, we have revised the conclusion paragraph.

“This study provides valuable insights on the trends of CAP hospitalizations over time using a large dataset of public hospitals and demonstrates the value of real-world data to inform public health interventions. By analyzing the incidence of CAP hospitalizations stratified by factors such as age group, sex or comorbidities our study provides relevant data to identify higher-risk populations. Our study also highlights the importance of having complete, accurate and timely health data from public and private health settings to inform policy making. Further studies are needed to evaluate the subsequent trends of CAP hospitalizations and its causes post COVID-19 to inform the design of strategies to mitigate the burden of community-acquired pneumonia.”

Congrats!

Thank you for your kind words and time dedicated to our manuscript. 

Response to comments from Reviewer 2

One significant limitation of the study is the exclusion of data from private hospitals, which could potentially affect the generalisability of the findings. If private hospitals experienced an increase in CAP hospitalisations during the study period, this could lead to an overestimation of the decrease in public hospitalisations, thereby distorting the overall trend of CAP incidence. Furthermore, the absence of private hospital data precludes a comprehensive understanding of the national hospitalisation trends. To what extent could the exclusion of private hospitals, where CAP hospitalisation rates may have differed from the public sector, influence the observed downward trend in CAP hospitalizations? How might the inclusion of private hospital data alter the study’s conclusions about the national burden of CAP?

We thank the Reviewer for the feedback to our paper and the opportunity to clarify this. We agree that not including private hospitals is indeed a limitation of our study that we had previously described in the discussion section. However, the majority of hospitalizations occur in public hospitals. In 2018, public and public-private partnership hospitals accounted for 82.9% of the hospitalizations of people aged 65 or older in mainland.[5] In our studied population, 81.1% of hospitalizations were from people in that age group (Table 1). To the best of our knowledge there is no database merging individual data for public and private hospitals nor there have been previous works in the country combining these sources. Following the reviewer’s suggestion, we have revised the methods, discussion, and conclusions accordingly. Revised sentences are included below and tracked in the revised version of the manuscript.

(methods) “In Portugal, public hospitals account for the majority of hospitalizations. In 2018, public and public-private partnership hospitals accounted for 83% of the hospitalizations of people aged 65 or older in mainland.[17]”

(discussion) “Fourthly, we were not able to include hospitalization data from private hospitals, which limited our capacity to fully capture the national trends of CAP hospitalizations. So, an increase in its share would lead us to overestimating the decrease of hospitalizations. However, it is unlikely this alone could explain the observed decrease, as the majority of hospitalization in Portugal occur in the public setting.[17] ”

(conclusions) “Our study also highlights the importance of having complete, accurate and timely health data from public and private health settings to inform policy making.”

The study encountered difficulties in aligning the hospitalisation data (numerator) with the total population affected by specific comorbidities (denominator). This mismatch could lead to either underestimations or overestimations in calculating the incidence rates of CAP associated with comorbidities. Such discrepancies may limit the precision of the study's estimates and its interpretation of the role of comorbid conditions in CAP hospitalizations. How would a more precise matching of the populations affected by comorbidities with the hospital admission data influence the observed incidence rates? What impact could this mismatch have on the interpretation of trends related to comorbid conditions such as diabetes, congestive heart failure, or chronic pulmonary disease?

Data to estimate the prevalence of CAP hospitalizations in people living with comorbidities had to be compiled from several sources (National Health Survey[6], Global Burden of Disease study[7], and EPICA study[8,9]). Hence, it was subject to the methods followed in each report (e.g., definition of age groups) and available years. We understand the reviewer’s comment, and have described in the limitation section of the discussion that this can lead to either an underestimation or overestimation. That is the case because, for some comorbidities, we considered people with 18+ in the numerator (hospitalizations) and people with 20+ in the denominator (prevalence), leading to underestimation (e.g., peripheral vascular disorders, HIV/AIDS, rheumatoid arthritis/ collagen vascular disease, solid tumor withot metastasis and metastatic cancer). In others, we considered people with 15+ in the denominator, leading to overestimation (e.g., complicated and uncomplicated diabetes, chronic renal disease/ renal failure and chronic pulmonary disease). Additionally, data for some comorbidities were available for “Portugal” and not for “mainland”. We rewrote the description of this limitation to inform readers that, for each comorbidity, there could be under or overestimation. We hope the revised version is clearer.

“Firstly, when calculating incidence rates per comorbidity, it was not possible to match entirely the definition of the population for which we observed the hospitalizations (numerator) and the number of people living with the condition (denominator), due to differences in age group and/or geographical area in available data. Their effect on incidence rates of each comorbidity is unclear, as these possible biases may have opposite directions and have simultaneously a risk of underestimation and overestimation”.

The absence of key data for certain years, especially the "present on admission" flag for the 2010-2012 period, introduces a potential bias in the estimation of CAP incidence. Without this flag, distinguishing between community-acquired and hospital-acquired pneumonia becomes impossible, which may lead to an overestimation of CAP-related hospitalizations. This limitation is particularly relevant for early years in the study, where the lack of this data could have inflated the reported incidence rates of CAP. To what extent does the absence of the "present on admission" flag during 2010-2012 affect the accuracy of the reported incidence rates for CAP hospitalizations? Could this limitation significantly impact the overall trend analysis, especially in the earlier years of the study?

Thank you for your comment. We agree that the lack of the "present on admission" flag in the earlier years could have inflated the reported incidence rates of CAP (cf. “Secondly, …” in paragraph about limitations). Results indicate that this has a limited impact in our results, as the overall trend seems well aligned before and after the inclusion of the present on admission flag (2013). Despite the limitation, we believe that including years 2010-12 in our study added valuable information for readers, as it allowed to study trends during a longer period and also to compare our results with a Portuguese study [1] with a partially overlapping period (2001-11).

The transition to the ICD-10-CM coding system in 2017 is another potential source of bias. While the study posits that the impact of this change was minimal, it is possible that the reclassification of cases under the new system could have affected the trend analysis, particularly if the coding criteria for CAP hospitalisations were altered. The lack of a comprehensive examination of this transition could limit the study's ability to account for variations in coding practices over time. How did the introduction of the ICD-10-CM coding system in 2017 influence the classification of CAP hospitalizations? Could this change have impacted the observed trends in CAP incidence, and to what degree should future studies adjust for potential shifts in coding practices?

While defining our study protocol, we performed an extensive review of the literature to identify lists of ICD-9-CM and ICD-10-CM codes that could be used to select the studied population. We aimed to reach a list that allowed us to compare CAP hospitalizations across the whole studied period and also with previous studies, but understand it may still be a limitation and revised its description to make it clearer. Despite this, we believe this limitation has a limited impact on our results, since the joinpoint analysis did not identify the first year with ICD-10-CM (2017) as a turning point. These results are aligned with a previous study comparing the frequency of Bacterial Pneumonia Admission Rate in Portugal (Prevention Quality Indicator 11) in different versions of the ICD, which found no impact of that change[10].

“Thirdly, ICD-10-CM was introduced in Portugal in 2017 (see S1 Table in Supplemental Material[18,19]), but the effect of this change also seems to be limited, as the joinpoint analysis did not identify 2017 as a turning point.”

We also agree with the Reviewer that it is important to provide information that can be easily found while conducting future studies. Thank you for bringing this up. We added to the discussion a mention to the supplementary material including the list of ICD-9-CM and ICD-10-CM codes and the references we had followed (previously only in the methods section – “Population”).

(Please see sentence above.)

Another notable limitation is the lack of data extending beyond 2018, preventing an analysis of CAP hospitalisation trends during and after the COVID-19 pandemic. Given that the pandemic has had a profound impact on healthcare systems worldwide, including shifts in the incidence and management of respiratory diseases such as CAP, the absence of post-2018 data leaves a significant gap in understanding the full impact of the pandemic on CAP hospitalisation trends. How did the COVID-19 pandemic influence CAP hospitalisation trends in Portugal, and what changes can be observed in the post-pandemic period? Would the observed trends in this study continue to hold in the context of the pandemic, or are they likely to be significantly altered?

We completely agree with the Reviewer that it would be important to include post-2018 data, particularly to study the tren

---

## [Decision Letter · Decision Letter 1]

25 Mar 2025

Characteristics and incidence trends of adults hospitalized with community-acquired pneumonia in Portugal, pre-pandemic

PONE-D-24-56883R1

Dear Dr. Joana Carneiro,

We’re pleased to inform you that your manuscript has been judged scientifically suitable for publication and will be formally accepted for publication once it meets all outstanding technical requirements.

Kind regards,

Alexandre Morais Nunes, Ph.D.

Academic Editor

PLOS ONE

Additional Editor Comments (optional):

Reviewers' comments:

Reviewer's Responses to Questions

**Comments to the Author**

1. If the authors have adequately addressed your comments raised in a previous round of review and you feel that this manuscript is now acceptable for publication, you may indicate that here to bypass the “Comments to the Author” section, enter your conflict of interest statement in the “Confidential to Editor” section, and submit your "Accept" recommendation.

Reviewer #1: All comments have been addressed

Reviewer #3: All comments have been addressed

2. Is the manuscript technically sound, and do the data support the conclusions?

Reviewer #1: Yes

Reviewer #3: Yes

3. Has the statistical analysis been performed appropriately and rigorously? 

Reviewer #1: N/A

Reviewer #3: Yes

4. Have the authors made all data underlying the findings in their manuscript fully available?

Reviewer #1: Yes

Reviewer #3: Yes

5. Is the manuscript presented in an intelligible fashion and written in standard English?

Reviewer #1: Yes

Reviewer #3: Yes

6. Review Comments to the Author

Reviewer #1: The paper presents an interesting and relevant topic.

The authors responded to each remark through detailed in my first revision.

Its a great article.

Reviewer #3: The discussion does touch on the exclusion of private hospital data, but it does not fully explore its potential impact on the observed downward trend or the overall burden of CAP. The review mentions that most hospitalisations occur in public hospitals and suggests that the exclusion of private hospital data might lead to an overestimation of the decline. However, the article review does not quantify this effect or consider how different hospitalisation criteria in private hospitals might influence the findings.

The discussion partially responds to the potential impact on trend interpretation for specific comorbidities. The review acknowledges the mismatch between hospitalisation data and population estimates, and does not explicitly discuss how this could influence trends related to conditions like diabetes, congestive heart failure, or chronic pulmonary disease.

The review does respond to the absence of the "present on admission" flag for 2010-2012, and explains that this may have led to an overestimation of incidence rates in those years due to the inability to distinguish hospital-acquired pneumonia. However, it also argues that the impact on overall trend analysis is likely limited because 2013 was not identified as a turning point. This directly addresses both the potential effect on accuracy and its significance for trend analysis.

The discussion does respond to the ICD-10-CM question. It mentions the introduction of the ICD-10-CM coding system in Portugal in 2017 and acknowledges the potential for coding changes to impact CAP hospitalisation classification. However, it argues that the effect of this change appears to be limited because the join point analysis did not identify 2017 as a turning point in the trends. While the discussion does not explicitly suggest adjustments for future studies, it implies that coding practices should be considered when interpreting trends.

The review does not fully respond to the CAP hospitalisation question. It acknowledges the limitation of not including data after 2018, which prevents analysis of CAP hospitalisation trends during the COVID-19 pandemic and post-pandemic periods. It does not provide any insights or hypotheses on how the pandemic might have influenced CAP trends or whether the observed trends would continue or change.

The evolution of mortality among hospitalised CAP patients was not explored in the study; it does not provide any analysis of how mortality rates evolved over time or how they might relate to the observed reduction in CAP hospitalisations. It does suggest that mortality trends could be an important area for future research but does not directly address the potential impact of including mortality data on public health strategies.

The discussion acknowledges the increased uptake of pneumococcal and influenza vaccinations during the study period and mentions previous studies that have linked vaccination uptake to a decrease in CAP hospitalisations. However, it also states that this specific relationship was not analysed in the study. The discussion suggests that future research linking individual-level vaccination data with hospitalisation records could provide further insight. While the study does not directly quantify the impact of vaccinations on CAP trends, it recognises the potential role of immunisation and the need for further investigation.

Although the authors have not considered all the suggestions, the study has some very meritorious points. The study is well conducted, the key points are well presented and the conclusions are well founded on the work carried out.

7. PLOS authors have the option to publish the peer review history of their article (what does this mean?). If published, this will include your full peer review and any attached files.

Reviewer #1: **Yes: **Andreia Matos

Reviewer #3: **Yes: **Ricardo de Moraes e Soares

---

## [Editor Report · Acceptance letter]

PONE-D-24-56883R1

PLOS ONE

Dear Dr. Carneiro,

I'm pleased to inform you that your manuscript has been deemed suitable for publication in PLOS ONE. Congratulations! Your manuscript is now being handed over to our production team.

Kind regards,

on behalf of

Professor Alexandre Morais Nunes

Academic Editor

PLOS ONE